# Intensification in Olive Growing Reduces Global Warming Potential under Both Integrated and Organic Farming

Salvatore Camposeo, Gaetano Alessandro Vivaldi, Giovanni Russo * and Francesca Maria Melucci *

Department of Agricultural and Environmental Sciences, University of Bari Aldo Moro, Via Amendola 165/a, 70126 Bari, Italy; salvatore.camposeo@uniba.it (S.C.); gaetano.vivaldi@uniba.it (G.A.V.)
* Correspondence: giovanni.russo@uniba.it (G.R.); francesca.melucci@uniba.it (F.M.M.)

**Abstract:** The relationship between agriculture and climate change is gaining prominence year by year in due to both adaptation and mitigation issues, because agriculture contributes to carbon emissions and acts as a carbon sink. Innovation on olive growing may help improve production systems for a more sustainable agriculture. In recent years, the olive sector is shifting towards intensification via a new growing system implementation with a strong economic impact. Indeed, the olive-growing systems are moving from low-density (<250 trees/ha) to medium-density (300–500 trees/ha), and mostly to super-high-density (>1200 trees/ha) systems. The aims of the present study were to compare these different olive-growing systems, managed by both integrated and organic farming, and to assess the effects of different agricultural practices on global warming potential (GWP), referring to one hectare and to one ton of olives as functional units. For both functional units and for all olive-growing systems, in the organic farming method, there is a greater environmental impact compared to integrated farming because of the higher number of mechanical operations (e.g., for weed control) in the former. The super-high-density growing system exhibited a lower GWP, considering both one hectare and one ton for both farming methods.

**Keywords:** olive growing; global warming potential; integrated farming; organic farming; super high density cropping system; climate change



## 1. Introduction

Climate change (CC) is currently causing an increase in earth mean temperature, with extreme weather events and spatially heterogeneous changes in rainfall and temperature patterns; these changes have prompted an enormous interest in the potential impact of CC and other components of global change on all living beings [1].

Growing greenhouse gas (GHG) emissions are altering the global atmospheric composition and they are strongly affecting climate stability [2]. The agri-food systems cause 21–37% of all greenhouse gas emissions, exceeding 5 Gt (a gigaton is one billion tons) of $CO_2$, 52% of which refers to cattle products [3]. Deforestation, use of fossil-fuel-based fertilizers, burning of biomass, food transport, and food waste are the main causes of GHG emissions sourced by agriculture [4]. There is a complex cause-and-effect relationship between agriculture and CC [5]: agriculture produces high quantities of greenhouse gases that cause CC [6]; however, agriculture has been seriously affected by CC, due to reduced productivity and increased exposure to risk of new pathogens [7]. The current global agri-food system is unsustainable and requires an agricultural revolution that is based on sustainable intensification and driven by sustainability and system innovation [8]. Particular attention needs to be paid to the agricultural sector in this context, because it also contributes to the environmental protection and provides a large range of "ecosystem services", as the United Nations Environment Programme (UNEP) and Millennium Ecosystem Assessment (MEA) have pointed out. The MEA distinguishes four types of ecosystem services, such as regulation of climate, air, and water quality, conservation of genetic diversity and biodiversity, and others [9].

Current food production systems failed to curb or reduce agricultural GHG emissions (+1.5% in 2018 vs 2017) so that more efficient, sustainable growing systems and farming methods are needed [3]. Sustainability has become the main paradigm of current policies, management practices, and research topics in many fields, particularly for European countries that are planning to reset the use of fossil fuels to zero by 2050; the agricultural sector will be affected by these changes [10]. In this context, the role of life cycle assessment (LCA) analysis [11], codified by ISO 14040–46 and applicable to processes, products, or services [12], allows the evaluation of the environmental loads through indexes that can guide the most compatible choices [13].

The carbon footprint (CF) is an environmental indicator that represents the amount of $CO_2$ emitted due to directly and indirectly associated total GHG emissions caused by a product, an organization, or a service [14]. Quantifying CF through LCA analysis allows the most impactful phases (called "hot-spots") of a process's life cycle to be observed and different products and different production methods of the same product can be compared. In this way it can be possible to redesign the production cycle and reduce GHG emissions from a sustainable perspective [15].

The global warming potential (GWP) is an indicator that quantifies the CF and describes the radiation-forcing impact of one mass-based unit of a given greenhouse gas related to an equivalent unit of carbon dioxide over the given period of time of 100 years ($GWP_{100}$), based on a relative scale that compares a specific GHG with an equivalent mass of $CO_2$, whose GWP, by definition, is equal to 1 [16]. The GWP is a measure of how much energy the emissions of one ton of a specific GHG will absorb and includes different types of GHGs (methane, nitrous oxide, hydrofluorocarbons, perfluorocarbons, sulphur hexafluoride, etc.).

Agriculture, though contributing to prosperity and health, causes many environmental impacts; if not sustainably managed, agricultural activities contribute to land degradation, natural resource exploitation and GHG emissions [2]. This is of extreme importance for widely cultivated species, where the impact will be extensive. In this regard, olive is one of the most cultivated fruit tree crops, covering more than 11 million ha (90% of the Mediterranean Basin), with Italy being the second largest producer in the world (more than 1.1 million ha). In this area, olive trees are an important element of the cultural heritage [17] and have a crucial role in the economy with significant social and environmental impacts as well [18]. Furthermore, extra virgin olive oil is the principal source of fat in the Mediterranean diet [10,18]. The Apulia region, with its 367,850 hectares of olive trees, represent the most important olive-growing areas in Italy (about 33% of the total Italian area) [19].

It has been well-stated that innovation on olive growing may help improve production systems for a more sustainable olive cultivation [20]. In recent years, the olive sector is shifting towards intensification by new growing system implementation, with a strong economic impact [21]. Indeed, the olive-growing systems are moving from traditional low-density (<250 trees/ha; LD) to intensive medium-density (300–500 trees/ha; MD), and mostly to super-high-density or super-intensive (>1200 trees/ha; SHD) systems [22]. LD systems represent 75% of the global olive surface and about 20% are intensive orchards [23]. The SHD growing systems originated in 1995 [24] and they cover about 5% of the world olive-growing areas [23]; a rapid expansion of SHD systems is taking place around both the Mediterranean basin and new areas, mainly due to their very high economic sustainability [25]; moreover, these new orchards are prone to precision agriculture applications [26].

The current study involved examining and analyzing the GHG emissions from different olive cropping systems and different farming methods widely used in the Apulia region. Making olive growing environmentally sustainable is the main challenge for the future but a sustainable intensification needs efficient orchard management [8]. In order to reduce GHG emissions from the olive oil sector, agricultural practices must be rigorously improved and waste production must be strongly reduced [3]. In a first step, the efficient use of agronomical inputs is required, particularly soil, water, and nutrient management, not renewable natural resources [27], and in a second step, the reuse of woody biomass is



necessary; depending on pruning management, the wood obtained by pruning is mainly used for domestic heating or shredded and buried to improve soil fertility, or picked and packed to be allocated for energy use within biomass power plants [28]. Woody residues from the disposal of olive groves can be chipped, turned into pellets and used in boilers, or transformed into compost; olive wood can be used as fuel for ovens in pizzerias and is suitable for cooking food.

However, intensification should be carefully evaluated from an environmental point of view. Pellegrini et al. [6] assessed the water footprint (WF) of different olive-growing systems, showing that MD and SHD systems have a lower WF than LD systems. GWP assessments focusing on agricultural sectors have been performed worldwide, while studies have not focused on different olive tree growing systems and farming methods. Indeed, up to now, olive-growing systems and farming methods have been studied from a more general environmental and/or economic point of view [29–33], focusing on different countries in the Mediterranean Basin, like Spain [34–36], Cyprus [37], Italy [15,38–41] and others [42–44]. In Italy, the distribution of LCA applications on GWP evaluation shows that energy is the most investigated field, followed by the construction industry, food, R&D, waste, and agriculture [45].

In order to cover this gap, we aimed for this study to quantify the impact of intensification in olive-growing systems in terms of GWP using LCA methodology. Our LCA proceeded in these three subsequent steps: evaluation of the three olive-growing systems, for both integrated and organic farmings; evaluation of the contribution of olive orchard management techniques for each growing system and each farming method; and evaluation of two end-of-life biomass reuse scenarios for MD and SHD systems: landfill for wood materials and incineration with energy recovery.

## 2. Materials and Methods

The current study involved the examination and analysis of three of the most representative drip-irrigated olive-growing systems in the Apulia region:

- Traditional or low-density systems (LD) with less than 250 trees/ha and olive groves more than 100 years old;
- Intensive or medium-density systems (MD) with tree density between 300 and 500 and olive trees between the ages of 30 and 50 years;
- Super-intensive or super-high-density systems (SHD) with over 1200 trees/ha and olive trees up to 20 years old.

Moreover, both integrated (INT) and organic (BIO) farming for each cropping system were considered.

### 2.1. Life Cycle Assessment (LCA)

The LCA multi-step procedure, as defined by the International Organization for Standards (ISO) series ISO 14040 (ISO, 2006) and ISO 14044 [13], was applied. According to those guidelines, the LCA methodology consists of four steps: (a) functional units and boundaries definitions; (b) life cycle inventory analysis; (c) life cycle impact assessment (LCIA); (d) results interpretation.

2.1.1. Functional Units and Boundaries Definitions

In this preliminary step, the functional unit and the boundaries of the studied system, such as assumptions, limits, and data quality requirements, are fixed.

A functional unit (FU) is a quantified description of the performance of the product systems used as a reference unit. To highlight the usefulness of the results obtained, two different FUs were considered:

1. 1 hectare of cultivated area (1 ha) to compare different olive-growing systems (LD, MD, SHD) and different agricultural practices;

2.  1 ton of fruits harvested (1 t) to compare different farming methods (INT, BIO) and as a reference point for any research developments beyond the farm-gate of this study (olive harvest).

The calculation of the GWP in reference to the unit of one hectare of soil allowed us to estimate the emissions due to all the processes and inputs necessary for the different studied theses. Compared to the unit of one ton of olives produced, the calculation of the GWP was more affected by the different productivities of the growing systems. According to Cerutti et al. in 2015, surface functional units are adopted when management practices are under evaluation.

The system boundaries (Figure 1) determine which unit processes to be included in the LCA study; they were fixed from a "cradle to farm gate" approach, that is, the olives harvest, in an area of 1 ha. The use of tractors and processing machines with their emissions, the plant protection products and fertilizers and their transport to the farm, and the end of life of the other materials used were also considered within the system boundaries.

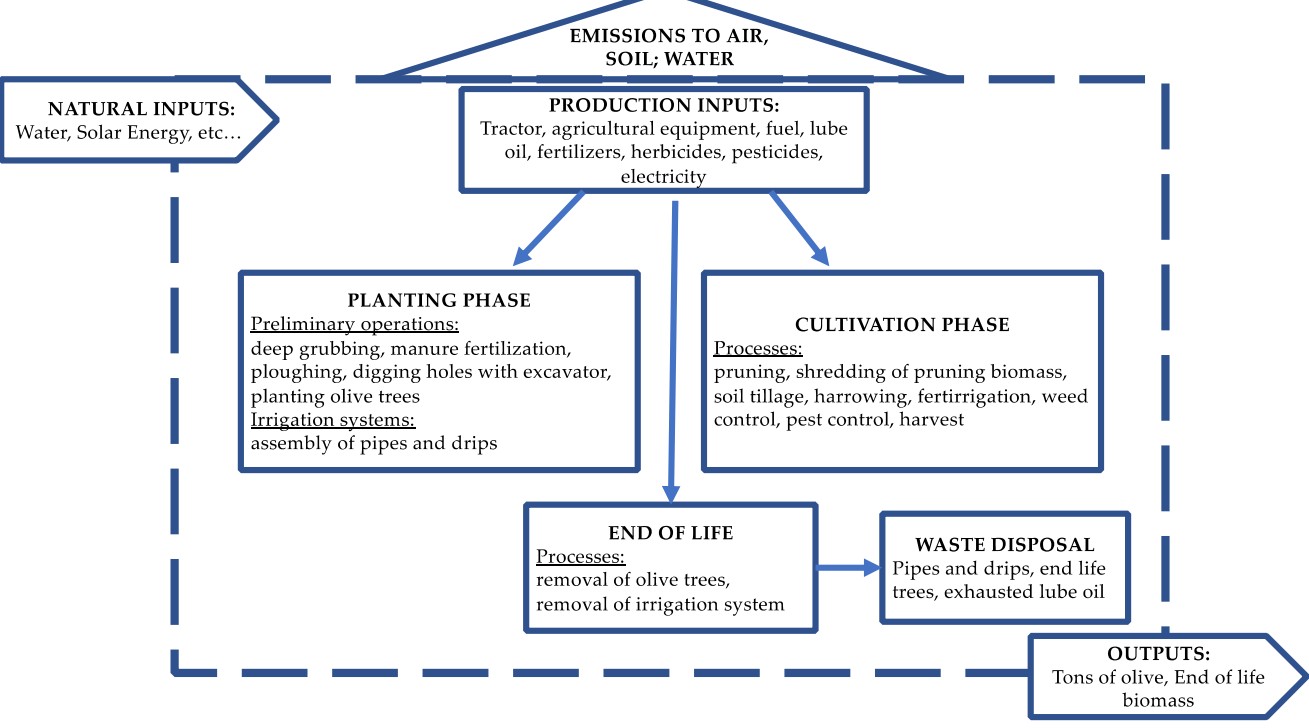

**Figure 1.** System boundaries of the process from the preliminary operations and the planting phase to the harvest of olives.

## 2.1.2. Life Cycle Inventory (LCI) Analysis

For the purpose of the study, data were taken for the whole orchard cycle, starting from planting olive trees till end of life. Primary data concerned the following items: orchard location, age of the trees, cultivar, growing system, farming method, productivity of olive orchard, irrigation volumes, amount of plastic pipe for irrigation, agricultural operations, agricultural machineries, and amount of pruning biomass (Table 1). These data were average values coming from the specific UNIBA (University of Bari, Bari, Italy) agronomical database on olive-growing systems including over 300 samples obtained during the last decade by on-farm surveys located in different Apulian olive-growing areas (Figure 2).

**Table 1.** Main agronomical characteristics and yields of the three olive-growing systems analyzed for both integrated (INT) and organic (BIO) farming methods: traditional systems (LD), intensive systems (MD), super intensive systems (SHD).

| Orchard Characteristics | LD INT | LD BIO | MD INT | MD BIO | SHD INT | SHD BIO |
|---|---|---|---|---|---|---|
| Cultivars | Cellina di Nardò, Ogliarola barese, Ogliarola salentina | | Coratina, Leccino, Peranzana | | Arbequina, Arbosana, Koroneiki, Lecciana ®, Oliana® | |
| Training form | Vase or globe | | Vase | | Central leader or Smartree ® | |
| Lifetime (years) | >100 | | 60 | | 20 | |
| Tree spacing (m × m) | 10 × 8 | 10 × 10 | 6 × 6 | 7 × 6 | 4 × 1.5 | 4 × 1.5 |
| Planting density (trees/ha) | 100 | 60 | 278 | 238 | 1666 | 1666 |
| Average annual productivity (t/ha) | 10 | 7 | 10 | 7 | 10 | 7 |
| Total productivity in 60 years (t/ha/60 yr) | 600 | 420 | 570 | 399 | 540 | 378 |
| Pruning method | Electric telescopic shears + elevator platform | | Telescopic pneumatic scissors | | Pruning machines | |
| Pruning biomass management | Shredded and used as soil mulching | | Shredded and used as soil mulching | | Shredded and used as soil mulching | |
| Harvesting method | Manual facilitated by electric/pneumatic comb or hook | | Mechanical by trunk shaker with reverse umbrella | | Mechanical by continuous straddle harvester | |

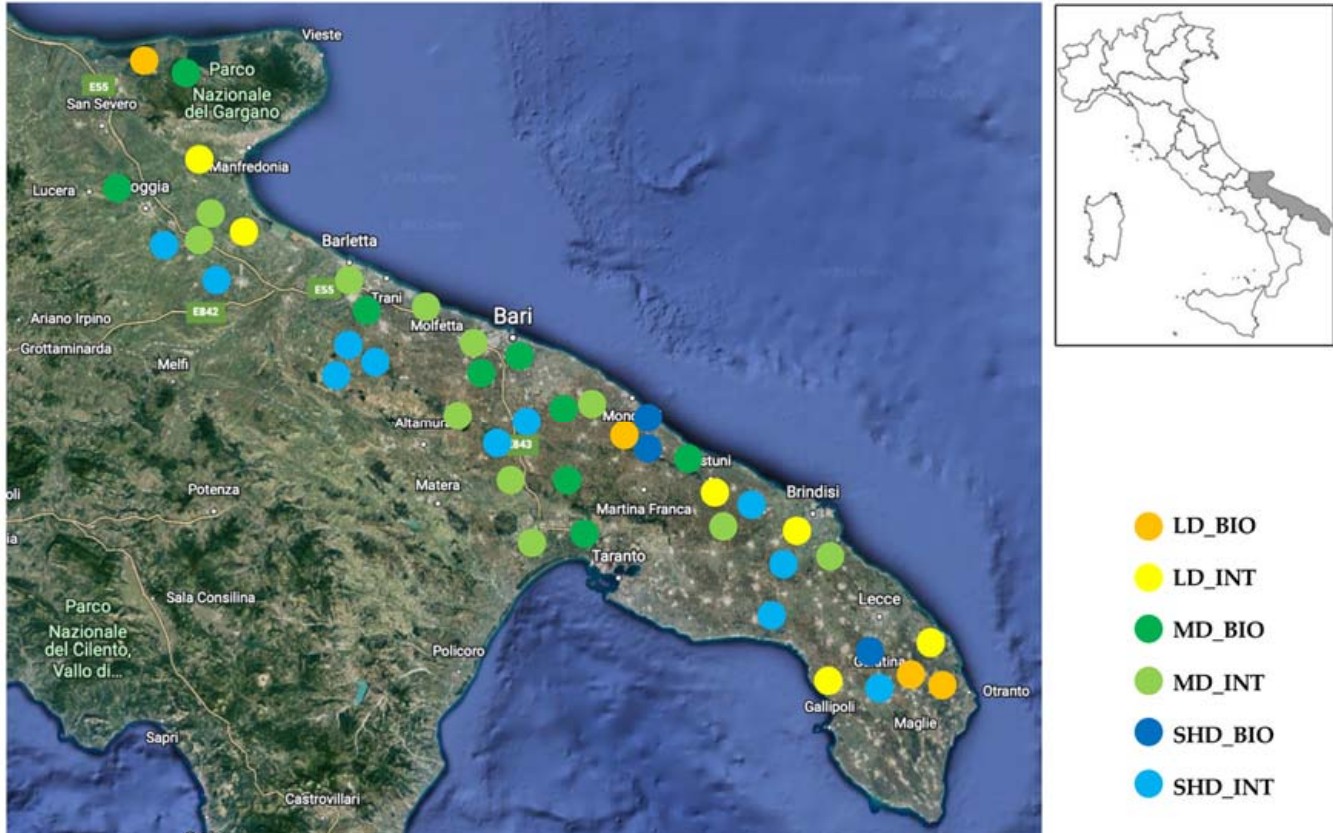

**Figure 2.** Map of the Apulia region with the location of the analyzed olive-growing systems.

Soils without anomalous physical and chemical characteristics were considered: a good amount of organic matter (1.0–2.5%), medium carbon to nitrogen ratio (9–12), and pH range of 6.8–7.5. Even if the dataset referred to the Apulian olive-growing conditions, this analysis could be extended to the other Mediterranean olive-growing areas [46–48].

Secondary data were found in the international literature and in the Ecoinvent 3.5 and GaBi databases to calculate materials and sources of energy used (fertilizers, weed control, pesticides, diesel, lube oil, electricity, water).

The production cycle of an olive grove lasts "n" years, beginning with its planting to the end of the production cycle (when there is no more economic sustainability). The lifetime of olive systems, or their production cycle called "useful life" (U.L.), changes depending on the growing system: over 100 years for the LD, about 60 years for the MD, and about 20 years for the SHD. Indeed, the intensification in fruit-tree-growing needs a reduction in tree size and strength: it leads to a strong reduction in juvenile phase and the economic lifetime of the orchard [49–52].

To perform the LCA analysis, a lifetime of 60 years was considered for each growing system. For the sake of simplicity, the production cycle of an olive grove was divided into four different phases:

- Planting, corresponding to the zero year;
- Juvenile unproductive, depending on growing systems;
- Transition, lasting about 2–3 years, and during these years the production is around half of the full production;
- Adult full production.

Productivity curves used for the different growing systems (LD, MD, and SHD) are reported in Figure 3.

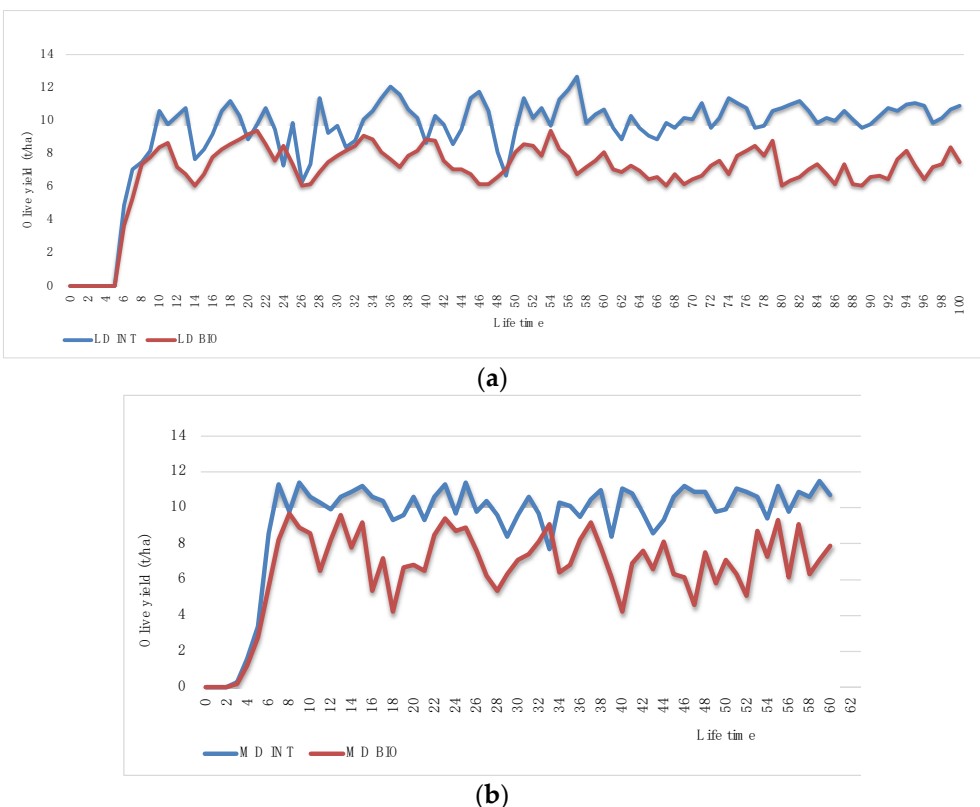

(a)

(b)

**Figure 3.** *Cont.*

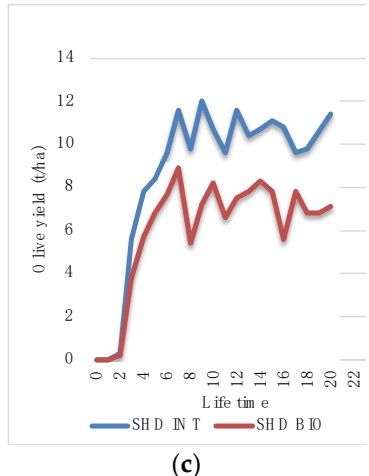

**(c)**

**Figure 3.** Productivity curve in terms of olive yield (t/ha) during the lifetime of the three growing systems for both integrated (INT) and organic (BIO) farming: (**a**) traditional systems (LD); (**b**) intensive systems (MD); (**c**) super intensive systems (SHD).

According to the average levels of yield for each olive-growing system, the productivity in 60 years for the LD system (the juvenile unproductive phase was not considered because the LD systems have a productive cycle longer when compared with the lifetime of 60 years), 57 years for the MD system (the first 3 years of the juvenile unproductive phase have not been considered), and 54 years for the SHD system (the first 2 years of the juvenile unproductive phase weren't considered and the production cycle has been multiplied by 3)) were compared to the two functional units (see Table 1).

The whole life cycle of the olive production process included:

- Extraction and treatment of raw materials;
- Seedlings production and manufacturing of machinery components, i.e., the agronomical inputs used;
- Transport of utilized input in the farm and from the farm to the field (transport of pesticides and fertilizers were overlooked because they were considered "local storehouse");
- Distribution of plant protection products and fertilizers, through atomizing and ploughing;
- Energy consumption of used machinery and their emissions;
- Reuse of biomass pruning (in particular, leaves and twigs cut up using mulcher and used as soil mulch; thicker branches sold as fuel);
- Final disposal of plastic materials (irrigation pipes) and end-of-life biomass (scenery);
- Land use and land occupation.

For each growing system, a lifetime of 60 years was considered to perform the LCA analysis because it allowed us to best represent all three crop cycles; the productive cycle of the SHD system was repeated 3 times.

Agricultural practices were grouped into different subcategories: soil preparation, irrigation, pruning, fertilizing, soil and pest management, harvest, and the end of life of the plant (scenarios).

Soil preparation is needed in order to prepare a solid agronomical base and improve soil fertility. The first stage was soil trenching in order to ease the development of plant roots, improving ploughing of the soil and water penetration. Following this was the soil fertilization and planting.

In soil preparation, fertilization was carried out by spreading liquid manure by means of a vacuum spreader. The olive tree seedlings were produced in the nursery.

The fertilization related to the preparation of the soil did not take into account the environmental load generated by the production of liquid manure and was considered 0 burden. The meat production chain originates numerous byproducts, such as animal

skin, slaughterhouse residues, litter, and liquid manure, which were diversified according to farming methods. The authors made this hypothesis to avoid the allocation of these byproducts for which it is difficult to obtain information in the Apulia region.

Only for the LD system, being centuries-old, soil preparation wasn't considered because the planting phase had a negligible contribution to the environmental impact compared to the cultivation phases. So, for the LCA analysis, this loses significance.

For the drip irrigation method, the quantities of pipes in high-density polyethylene (considering the diameter, weight, length, and life cycle of the pipes, according to the dimension of 1 ha) were calculated. In the SHD system, the largest quantity of irrigation pipes was used (2600 m) because there were more implant rows in 1 ha compared to the MD (1760 m) and the LD (1160 m) systems.

Pruning was completed using different methods, according to each system [22,46]. In the SHD system, manual and mechanical pruning should be done at the planting phase; in the MD system, for the first 3 years, manual pruning was used. In the LD system, a light pruning was performed every five years with electric telescopic shears. To facilitate the pruning, the operator gets close to the crown (4–5 m high) using an elevator platform. The pruning time was about 25 min per tree. In the MD system, pruning was done every 2 years, with a saw and telescopic pneumatic scissors (the height is about 2.5 m). Pneumatic tools were operated by an air compressor powered by tractor power take off (PTO). The pruning time was about 15 min per tree. In the SHD system, manual and mechanical pruning was done by pruning machines with a cutter bar or disk that cut the upper part (topping) or the side (edging) of the crowns; to eliminate the basal branches trimming was used. The pruning time was 2 h per hectare. The pruning biomass (leaves and twigs) was shredded and used as soil mulching in each system.

The pruning biomass (kg/ha) produced by different cultural systems (Figure 4) was higher in LD systems, with a production of 400 kg/ha, 40% of which was wood and 60% was leaves. In the MD systems, the average of the pruning biomass was 60 kg/per tree, 20% of which was wood and 80% was leaves. Lastly, in the SHD systems, pruning biomass was about 2,5 kg/per tree and there were leaves and shoots.

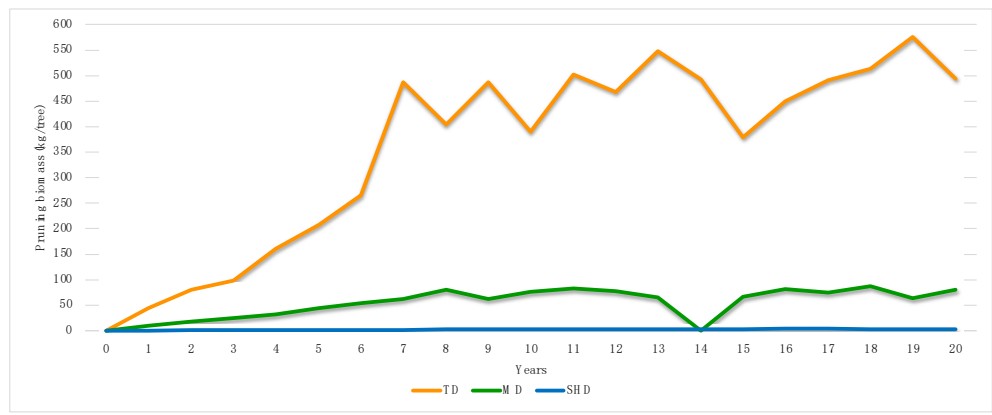

**Figure 4.** Pruning biomass (kg/tree) from the traditional (LD), intensive (MD), and super-intensive (SHD) growing systems.

The amount of fertilizer was applied following the current Disciplinary for Integrated Production (DIP) in the Apulia region (www.regione.puglia.it, (accessed on 18 March 2022), which states maximum amounts (kg/ha) for macronutrients according to different levels of olive yields; the data used are shown in the Supplementary Materials Tables S1 and S2). A standard production of 6–10 t/ha of olives was evaluated.

To evaluate the organic fertilizers, the values of organic fertilizers present in Agribalyse [53] were used; considering a release of $CO_2$ equal to 1941 kg $CO_2$ eq/t of organic fertilizers, and from the system, the loads avoided by chemical fertilization were deducted.

For fertilizers outside of the database, an NPK calculator that allows consideration of doses for nutrients of the desired fertilizer was used.

Regarding pest control, the main pests present in the Apulia region were considered. The number of treatments and the amounts of active substances per hectare were applied following the Integrated Pest Management method for INT cases and Regulation (EU) 2018/848 of the European Parliament [54] on organic agriculture for BIO cases (Table S3).

The emissions in the production phase were only considered for the plant protection products used because the toxicity testing is not necessary in the CF analyses. To reproduce this in the GaBi software, the fungicide "copper oxychloride" and the "Fungicides, at regional Storehouse" process for a generic fungicide was used.

The *Bacillus thurigensis* subsp. *Kurstaki* is produced by the submerged fermentation on liquid artificial substrates; as this process is not included in the GaBi database, a generic yeast ("Yeast Past") was used. For the natural pesticides Spinosad, the process "Yeast Past" was considered; while, for the mineral oil, the "Paraffin" process was used.

Harvest is completed using different methods according to the olive-growing system. Mechanical harvesting with electric/pneumatic comb or hook (1200 beats/min and 100 WP) used by an operator is involved in the LD system; the operator is facilitated by an elevator platform. The harvest time is about 25 min/tree [48]. The trunk shaker with reverse umbrella was used in the MD systems; it is a harvest machine (65 kW power) that has a hydraulic arm shaped as a reverse umbrella. The function time of this machine is about 15 min/tree. In the SHD systems, straddle-harvesting machines were used (diesel consumption 15.48 L/ha) [55], which takes 2 h/ha to harvest. These data have been found on the production companies' websites and their technical sheets.

In the LCA analysis, for agricultural machinery used, the roof shelter was also considered. The machines that are not included in the databases (for example, straddle-harvesting machine, flail mower, and elevator and pruning machines for the SHD) and their emissions were reproduced by a "General Agricultural Machinery" process. The tractor emissions were used by modifying them according to the specific machinery operation and by consulting the production company's website and technical sheets (weight and consumption/per working hour).

We considered the transport distance to local retailers by lorry to be about 30 km for the distribution of fertilizers and pesticides as well as for irrigation pipes and the elevator. Furthermore, in the scenarios with the incineration of end-of-life biomass, a transport distance of 30 km was considered. Finally, we considered the transport distance from the field to the farm to be 5 km for the used materials (such as fertilizers, pesticides, end-of-life biomass, etc.), using a truck and trailer process.

Using the LCA analysis, in addition to olive production and 1 ha of cultivated soil, the following coproducts were obtained: (1) annual pruning biomass, partly sold and partly used as soil mulching; (2) plastic materials and irrigation pipes replacement; and (3) end-of-life tree removal biomass.

End-of-life biomass resulting from olive tree removal depends on different factors (cv, pedoclimatic environment, rain-fall rate, etc.). In the LD systems, the end of life has been overlooked because the trees' life cycle is more than 100 years.

Meanwhile, in the MD and SHD systems, in the absence of validated data, we considered the following quantities acquired by the farms from which primary data have been taken: 600 kg/tree of end-of-life biomass for the MD system and 200 kg/tree of end-of-life biomass for the SHD system. In the "standard" end-of-life plan, the biomass reuse was not assumed.

### 2.1.3. Life Cycle Impact Assessment

According to the Kyoto Protocol there are six main GHGs: carbon dioxide ($CO_2$), methane ($CH_4$), nitrous oxide ($N_2O$), hydrofluorocarbons (HFCs), perfluorocarbons (PFCs), and sulphur hexafluoride (SF6). For each GHG, a GWP has been calculated to reflect how long it remains in the atmosphere, on average, and how strongly it absorbs energy. Gases

with a higher GWP absorb more energy, per pound, than gases with a lower GWP, thus contributing more warming. The GWP was developed to allow comparisons of the global warming impacts of these different gases.

GaBi 9.2 software was used to assess the environmental impact of the analyzed systems. For both of the two FUs (1 hectare of cultivated area and 1 ton of olives harvested), analyses were conducted using the CML 2001 method [56,57].

The GWP values, expressed as kg $CO_2$ equivalent, was calculated over a period of 100 years ($GWP_{100}$). The GWP is related to one hectare of cultivated area and to one ton of olives harvested.

## 3. Results

As mentioned in the introduction, the study was carried out to examine the sustainability, in terms of carbon footprint, of three different olive-growing systems: the traditional system (LD), the intensive system (MD) and the super-high-density system (SHD).

In this section, the results of the environmental impacts from the cradle-to-gate systems using 1 hectare of cultivated area and 1 ton of olives as the FUs are presented first, then the results of the comparison between different agricultural practices are shown.

### 3.1. $GWP_{100}$ per ha of Cultivated Area

Considering 1 ha as a FU for calculating the $GWP_{100}$, it appears that for all growing systems, the organic farming method showed a greater impact than integrated farming; in particular (Figure 5):

- The LD_BIO system has the highest level of GHG emissions ($266 \times 10^3$ kg $CO_2$ eq/ha), followed by the MD_BIO system ($229 \times 10^3$ kg $CO_2$ eq/ha);
- For organic farming, the SHD_BIO system is the one with the lowest CF ($200 \times 10^3$ kg $CO_2$ eq/ha) compared to LD_BIO and MD_BIO;
- The SHD_INT system has the smallest environmental impact ($132 \times 10^3$ kg $CO_2$ eq/ha)

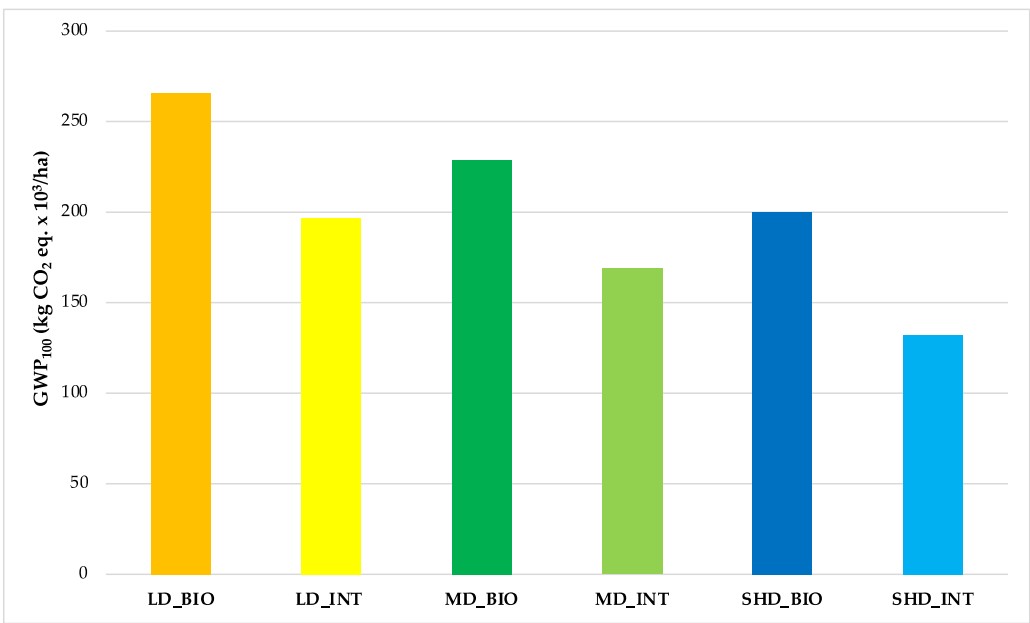

**Figure 5.** (FU: 1 ha) Value of global warming potential per hectare for the three olive-growing systems in both integrated (INT) and organic (BIO) farming methods: traditional systems (LD), intensive systems (MD), super-intensive systems (SHD).

In particular, in comparing organic farms, it was clear that LD_BIO was the most impactful system with an emission of $266 \times 10^3$ kg $CO_2$ eq/ha. In addition, for the integrated farming, LD systems show the highest GHG emissions ($197 \times 10^3$ kg $CO_2$ eq/ha).

### 3.2. GWP$_{100}$ per Ton of Olives

Results remain unchanged when it is considered that olives yield the FU (1 ton), because organic farming has a higher GWP$_{100}$ regardless of the growing system; in fact, in Figure 6, it can be observed that the LD_BIO system shows higher values, followed by the MD_BIO system, and the SHD_BIO system with the lowest emissions.

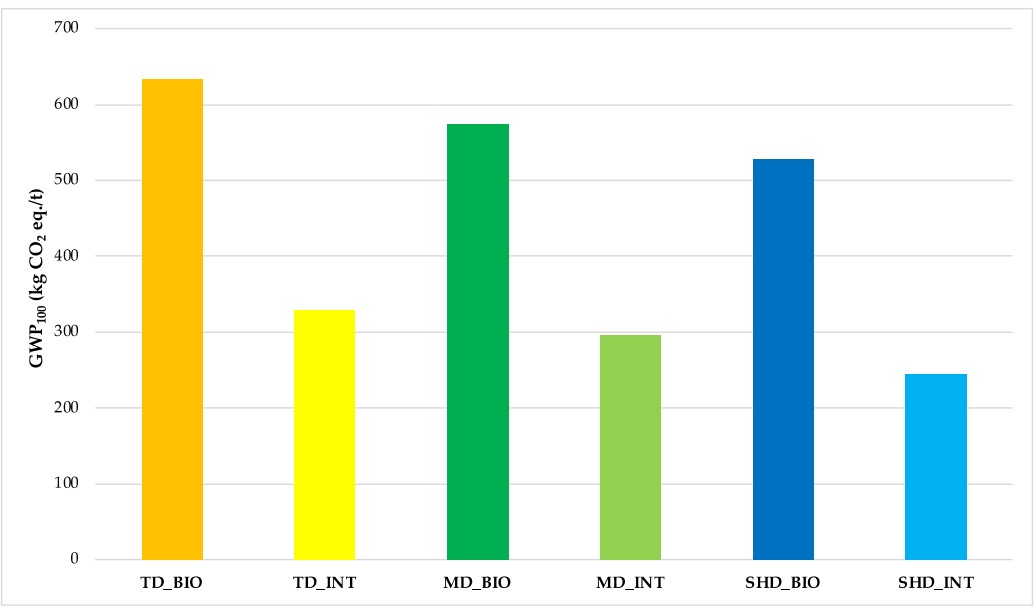

**Figure 6.** (FU: 1 ton) Value of GWP$_{100}$/ton of olive for the three olive-growing systems in both integrated (INT) and organic (BIO) farming methods: traditional systems (LD), intensive systems (MD), super-intensive systems (SHD).

Comparing the values of the GWP$_{100}$ in farms following the organic method, it has been observed that SHD_BIO is the system with the lowest CF (529.1 kg $CO_2$ eq/t) against LD_BIO at 633.33 kg $CO_2$ eq/t.

The same situation happens when comparing farms following the integrated method: the LD_INT system has the higher value of GWP$_{100}$ (328.33 kg $CO_2$ eq/t) followed by 296.49 kg $CO_2$ eq/t for MD_INT and 244.44 kg $CO_2$ eq/t for SHD_INT.

### 3.3. GWP of Agricultural Practices in Different Olive-Growing Systems

The figures below (Figures 7–9) report the percentage of each agricultural operation in the different olive-growing systems and farming methods with their weight on the GWP index (considering 1 ha as FU).

In particular, for the LD_INT system (Figure 7a), the highest rate is the one relating to pruning (42.5%, because there is a large amount of pruning biomass) and harvest (25.1%); this is caused by the high number of working hours of the elevator to assist the user during those operations.

Meanwhile, in the LD_BIO system (Figure 7b), where the percentages of each cultural practice are to follow, the rate of fertilizers (38.4 %) is higher than in the LD_INT system (19.7%).

The rate of pesticides and weed control is more relevant in LD_BIO (8.2%) in relation to the rate of this operation in LD_INT (6.4%) due to the increased number of mechanical operations required for weed control.

In the MD_INT system (Figure 8a), the greater environmental impact is caused by pruning (48.3 %), while in the MD_BIO (Figure 8b) system, the fertilizers have a slightly greater impact (43.6 %).

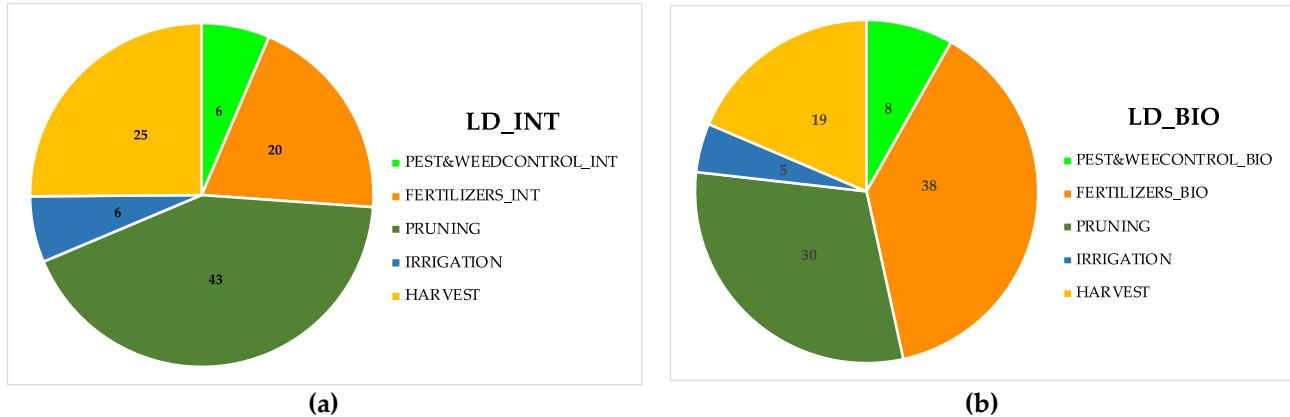

**Figure 7.** Orchard management contribution to GWP for traditional systems (LD), in integrated (INT, **a**) and organic farming (BIO, **b**).

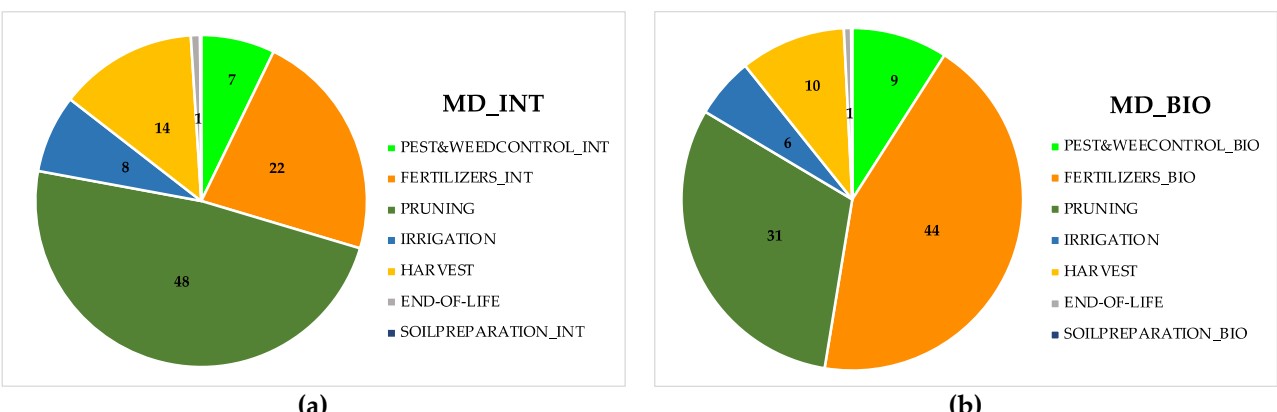

**Figure 8.** Orchard management contribution to GWP for intensive systems (MD) in integrated (INT, **a**) and organic farming (BIO, **b**).

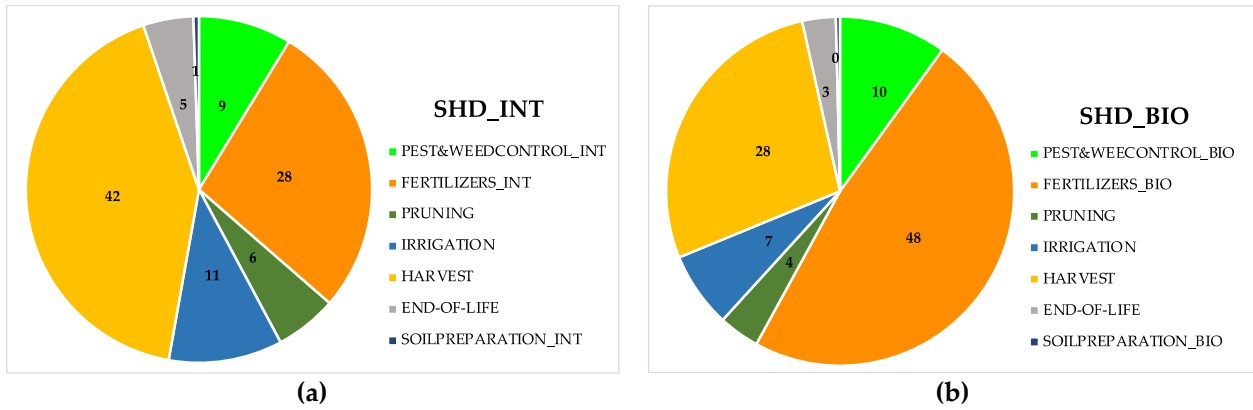

**Figure 9.** Orchard management contribution to GWP for super-intensive systems (SHD) in integrated (INT, **a**) and organic farming (BIO, **b**).

Finally, the most impactful phase for the SHD_INT system (Figure 9a) is the harvest (41.9%), followed by fertilizers (27.6%); meanwhile, in the SHD_BIO system (Figure 9b), the fertilizer process accounts for 48% of the GWP.

By comparing the integrated and organic farming methods in the three olive-growing systems, it can be seen that:

- The operation of pest and weed control has a slightly higher environmental load in organic farming and in the SHD system;

- Fertilizer management has a greater impact on organic farming;
- Pruning has a greater impact on the GWP in the LD and MD systems, while in the SHD system, mechanical pruning has a small impact;
- The olive harvest operation contributes 42% to the GWP of the SHD_INT system.

### 3.4. Scenarios for the End-of-Life Biomass of Olive Trees

Figure 10 represents the emissions of $CO_2$ due to the woody biomass from the removal of olive trees at their end of life.

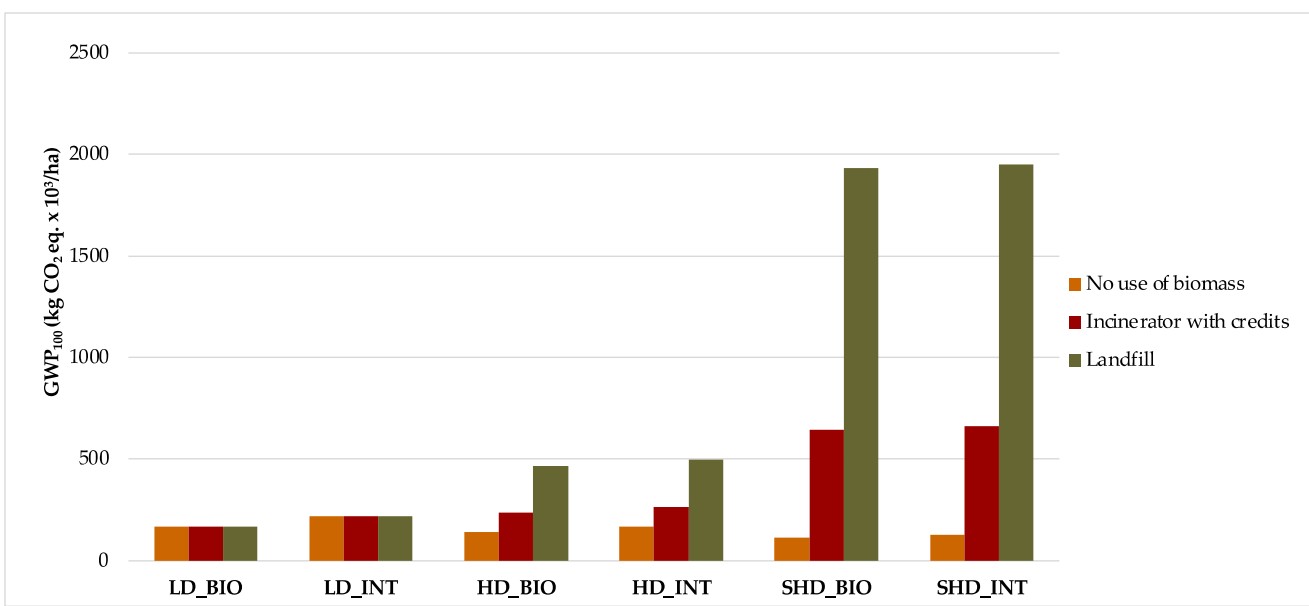

**Figure 10.** Different scenarios for the end-of-life biomass of the olive trees in the different combinations of olive growing × farming method.

For LD systems, there is no removal of trees while for the MD, there is only one removal of trees and for the SHD there are three removals of trees, considering 60 years as the lifetime. By analyzing scenarios, the emissions are increased: 15 times for the landfill where biogas is used and 3 times for the incineration with credits. Figure 10 shows that the landfill waste disposal induces greater environmental impacts on CC while the incineration with energy recovery has a low $GWP_{100}$.

### 4. Discussion

This is the first LCA study on the comparison among three different olive-growing systems (traditional system, LD; intensive system, MD; super-high-density system, SHD) in integrated (INT) and organic (BIO) farming methods.

Many researchers have analyzed systems, processes, and products through LCA and CF approaches (Table 2). The major differences between the farm cases of the present study and the studies reported in the scientific literature are in the consideration of very different productivities and different scenarios of woody biomass disposal, so a comparison of the results is difficult.

Data have been extrapolated over time intervals of 60 years of similar bibliographical studies in order to compare the results obtained in the present study, even if the lifetime values in the bibliography are lower. In addition, there are very different datasets, such as the lack of irrigation, the carbon stock of other crops carried out to reduce fertilization, fertilization, and productivity per hectare.

**Table 2.** Comparison with the results of other authors.

| Authors | Software | F.U. | Unit | Quantity | Notes |
|---|---|---|---|---|---|
| De Gennaro et al. (2012) [29] | SimaPro | 1 ton | kg $CO_2$ eq/t | 542.2 (MD) 707.1 (SHD) | Lifetime 48 years MD: 400 trees/ha; 11 t/ha SHD: 1667 trees/ha; 9 t/ha Drip irrigation Integrated management |
| Salomone and Ioppolo (2012) [15] | SimaPro 7.2 | 1 ton | kg $CO_2$ eq/t | 481 (conventional dry farming) 614 (organic irrigated farming) | Organic and conventional irrigated and dry farming 4 t/ha |
| Proietti S. et al. (2014) [38] | SimaPro 7.1 | 1 ha | t $CO_2$ eq/ha /year | 1.507 ($\times 60y = 0.90 \times 10^5$) | cv Leccino: $5.5 \times 5.5$ m Carbon stock Lifetime 11 years Integrated management |
| Proietti P. et al. (2016) [39] | SimaPro 8.0.3.14 | 1 ha | t $CO_2$ eq/ha /year | 1.837 ($\times 60y = 1.1 \times 10^5$) | Lifetime 14 years MD olive-growing system Net $CO_2$ sequestration cv Leccino: $5.5 \times 5.5$ m; 330 trees/ha Integrated management |
| Romero-Gamez et al. (2017) [35] | SimaPro 8.0.4.30 | 1 ton | kg $CO_2$ eq/t | 178 (LD, organic) 216 (LD, integrated) 309 (MD, integrated) 308 (SHD, integrated) | Irrigated olive growing LD: 100 trees/ha; 6 t/ha MD: 250 trees/ha; 10 t/ha SHD: 1900 trees/ha; 12 t/ha |
| De Luca et al. (2018) [40] | SimaPro | 1 ha | kg $CO_2$ eq /ha/50 yr | $3.65 \times 10^5$ (LD, conventional) $3.60 \times 10^5$ (No-tillage, low chemical) $3.82 \times 10^5$ (Zero chemical) | Lifetime 50 years |

Considering 1 ha as a functional unit and tolerating the differences between lifetime adopted, and considering them extended to the period of 60 years, the CF of the LD_BIO system is lower ($266 \times 10^3$ kg $CO_2$ eq/ha) than the values obtained by De Luca et al. [40] ($365 \times 10^3$ kg $CO_2$ eq/ha/yr). This difference can be attributed to the different amounts of fertilizer used and to the herbicides used for weed control.

In the present study, it was found that the CF of the LD_INT system is $1.97 \times 10^5$ kg $CO_2$ eq/ha, which is higher than the results obtained by Proietti et al. [38] ($90 \times 10^3$ kg $CO_2$ eq/ha/year) and lower than De Luca et al. [40] ($365 \times 10^3$ kg $CO_2$ eq/ha/yr). The difference with Proietti et al. [38] can be attributed to the calculation of carbon and annual $CO_2$ eq. stocked in the olive grove during its life cycle.

Considering 1 ton/ha of olives grown and comparing the CF of LD_INT (328.33 kg $CO_2$ eq/t) in the current study with the value obtained by Salomone and Ioppolo [15] (481 kg $CO_2$ eq/t) and the value obtained by Romero-Gamez et al. [35] (216 kg $CO_2$ eq/t), a comparison shows that the value of the present study is lower than the first and higher than the second. For these two papers, the differences found are because of the uncertainty of productivity due to the cultivated species.

In this study, the value of the CF for the LD_BIO system has been estimated at 633.33 kg $CO_2$ eq/t, which can be compared to the values obtained by Salomone and Ioppolo [15] equal to 614 kg $CO_2$ eq/t and by Romero-Gamez et al. [35] equal to 178 kg $CO_2$ eq/t. This very low value is because of the low amount of fertilizer used and the use of leaf treatments.

The CF of the MD_INT system is estimated at 296.49 kg $CO_2$ eq/t and can be compared with the value of De Gennaro et al. [29] (542.2 kg $CO_2$ eq/t) and the value of Romero-Gamez et al. [35] equal to 309 kg $CO_2$ eq/t. In the first paper, the amount of fer-

tilizer used is not in agreement with the doses present in the Disciplinary for Integrated Production (DIP) for the Apulia region while, on the other hand, there is substantial agreement with the data of the second paper.

According to De Gennaro et al. [29] and Romero-Gamez et al. [35], the CF of the SHD_INT system is equal to 707.1 kg $CO_2$ eq/t and 308 kg $CO_2$ eq/t, respectively, while in this study, the value of the CF of the SHD_INT system is lower, equal to 244 kg $CO_2$ eq/t. For the first paper, the major differences are due to the irrigation volumes and the productivity of the olive grove and with the second paper, the major differences are due to the count of $CO_2$ stored in the biomass of the olive grove.

Despite the diversity of agronomical data, climate data, and system boundaries considered, it is observed that the variability of data is included, for both FU, in only one order of magnitude of the results, sometimes with very similar numerical values.

The main findings of the study are:

1.  In farms with the same farming method (especially INT or BIO), the greater environmental impact is that of the LD, followed by the MD and the SHD;
2.  Considering 1 ha as a functional unit, it can be seen that organic farms have a higher $GWP_{100}$ compared to integrated farms; particularly, among the TD cropping systems, the BIO farming shows + 57,45% of GHG emissions compared to the INT farming; in the same way, the MD_BIO shows + 57,54% of emissions compared to MD_INT; and SHD_BIO shows + 60,24 % of emissions when compared to SHD_INT, in accordance with the study by Clark S. [58].

The LCA analysis also emphasized that there are aspects of the olive production chain that have the greatest impact on overall carbon footprint. Indeed, the agricultural practices have a lot of influence on GHG emissions. In fact, the agricultural practices play an important role in GHG emissions; indeed, optimization of olive agricultural practices, such as soil management [59] and fertilization [60], has the potential to decrease atmospheric carbon either by reducing emissions or enhancing sequestration processes in the soil. Although organic fertilizers have a lower incidence on the GWP, the greater impact on the GWP of fertilizer depends on the mechanical operations and the largest number of treatments.

To reduce the environmental impact associated with olive farms, it need to act on the high levels of mechanization (pruning and harvest) and the consequent use of fossil fuels, and on the use of pesticides and fertilizers.

Results obtained with the application of the LCA methodology in the Apulian olive-growing systems allow us to make suggestions about two different aspects:

- How to design a more efficient and environmentally friendly olive orchard;
- How to use LCA analysis as a method to underline hot-spots in the orchard management.

End-of-life biomass can also be used in biomass reactors with biogas production and renewable energies (and non-fossil sources) or could be used in composting. However, these scenarios are difficult to estimate because of the combustion efficiency (for the boiler) and because of the chemical reactions under anaerobic conditions (for biogas fermentation tank). Further details are needed, although it is underlined how woody biomass disposal is a key point in the olive-growing chain.

Land management activities offer a potential for both reducing GHG emissions and also for sequestering carbon. Possible mitigation measures include all actions that can reduce emissions or increase removals of GHGs, particularly $CO_2$ emissions related to changes in carbon stock in soils and biomass.

To reduce the environmental impact caused by soil management and the use of pesticides, electric tractors using electricity from renewable sources or biomethane self-produced by biogas reactors could be used.

In addition, to improve the CF of the fertilizers, one possible approach would be to use green manure to cultivate legumes, spread livestock wastewater, or use treated municipal wastewater.

## 5. Conclusions

Our study, based on a very large temporal and spatial dataset, referred to the Apulia region, one of the most important Mediterranean olive-growing areas, shows that, in implementing agricultural intensification, it is now possible to combine the modern agriculture with a high standard of environmental performance [61], first of all improving the use efficiency of natural resources.

Furthermore, our research suggests that, to mitigate and decrease the carbon footprint of olive production, it is necessary to optimize all the agricultural practices that have high levels of GHG emissions.

Paradoxically, it is the high tree density that represents the essential reason for the environmental sustainability of the new growing systems, which reduce the olive water footprint as well [6]. Sustainable intensification can embody the concept of "producing more with less", so that super-intensive olive groves should be renamed as super-sustainable olive groves.

This information, combined with other key aspects, such as economic, landscaping, and social considerations, could provide a starting point for the formulation of guidelines for both the rational management of the olive orchard and for regional, national, and community agricultural policies, considering that the new planting systems are prone to precision agriculture applications as well.

These results open up a number of avenues for future research, particularly focusing on soil management that increases the carbon sequestration or on carbon sinks [62]. Further works are needed to determine the toxicity of fertilizers and treatments used for disease control and to assess the cropping systems under rainfed conditions and in the case of unconventional wastewater reuse.

**Supplementary Materials:** The following supporting information can be downloaded at: https://www.mdpi.com/article/10.3390/su14116389/s1. Table S1. Standard amounts of macronutrients for the juvenile unproductive phase fertilization of olive growing, according to the DIP of Apulia region, both for the integrated and organic management. Table S2. Standard amounts of macronutrients for the adult phase fertilization of olive growing, according to the DIP of Apulia region. Table S3. Control of the most common diseases, phytophagous and weed in Apulia region, both for INT and for BIO farmin.

**Author Contributions:** Conceptualization, S.C. and F.M.M.; methodology, G.R. and F.M.M.; software, G.R. and F.M.M.; validation, G.A.V., S.C., G.R. and F.M.M.; formal analysis, S.C. and G.R.; investigation, F.M.M. and G.A.V.; resources, S.C. and G.R.; data curation, G.A.V., F.M.M.; writing—original draft preparation, F.M.M.; writing—review and editing, F.M.M., S.C. and G.R.; visualization, G.R.; supervision, S.C. and G.R.; project administration, S.C. All authors have read and agreed to the published version of the manuscript.

**Funding:** This research received no external funding.

**Institutional Review Board Statement:** Not applicable.

**Data Availability Statement:** Not applicable.

**Conflicts of Interest:** The authors declare no conflict of interest.

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
