# Peer review of "Intensification in Olive Growing Reduces Global Warming Potential under Both Integrated and Organic Farming"

_sustainability, doi:10.3390/su14116389_

Round 1
Reviewer 1 Report
The paper is sufficiently well written and provides some useful information. I have few concerns. First, the authors have used data from sources other than their own and in that sense all of the data are secondary. I would be concerned about the quality of these data. Second, the authors present results from their analyses of these data, but they do not provide any further explanations as to why these results. As a result, the sections of 'discussion' and 'conclusions' are weak.
Some other observations are following.
Introduction section
Line 21: Correct spelling of ‘both’.
Introduction is little too long. Consider reducing the section; for example: is the paragraph lines 52-66 is necessary?
Line 81: English composition: Agriculture …. Causes.
It is better to provide the reader the research goals and the main approach to achieving the goals in the beginning of the Introduction section rather than at the end.
Figure 1 is hard to read, especially the text in light color.
Materials and Methods
Lines 178-188: I am confused with the use of the term ‘Primary data’. It is commonly used to mean the data generated by the research being described. But here they are average values derived from other studies and sources. Please make this clear to the reader.
Lines 193-197: Why is the production cycle low (20 years) for the SHD systems? Why does it increases with lower density? Brief explanation would be helpful.
Line 209: Table 2 is referred here, but I think it is missing in the paper.
Lines 360-367: Why are the GHG emissions higher in case of LD systems as compared to the MD and SHD systems? What is the logical explanation?
Reviewer 2 Report
In general the paper written in good structure. Some weakness points are listed as follow:
Introduction
1- Page 1 Line 31: Please check the CC abbreviation. Also, correct “Earth” to “earth”.
2- Page 1 Line 37: Please define “5 Gt” for reader
3-Page 2 line 47: as the UNEP (United Nations Environment Programme) and MEA (Millennium 48 Ecosystem Assessment) have pointed out. Corrected to the United Nations Environment Programme (UNEP) and Millennium 48 Ecosystem Assessment (MEA).
4- More explanations are required to clarify the research gap of this study.
Materials and Methods
5- The resolution of figure 1 is very weak and unreadable. I recommended to re-draw with readable colors.
6-Page 5 Line 138: Please define the “UNIBA”.
7-Figure 2 and Figure 3: I recommended to change the x-axis from years to Life time or age.
Results
8- Page 10 line 362: More information are required on the calculations ore measurement method of GHG emissions.
9- I recommended to write the conclusion part in separate section.
Reviewer 3 Report
The authors have studied the Intensification in olive growing reduces Global Warming Potential under both integrated and organic farming method. Many results, but extensive improvements must be done before considering it for publication. My comments are detailed bellow, as follows:
Aspect of the manuscript must be improved according to the Instructions for authors: text justified, without interspaces between the paragraphs of the same section.
Keywords must reflect the main characteristic words of the paper (usually reflected also by the title) in the best way to increase the paper's relevance and chances to be find when searching it after key words. So, for the actual title, I suggest the following keywords: olive growing; Global Warming Potential; integrated method; organic farming method; climate changes.
No needing numbering the aims of the study (L132-137). Just separate them by semicolon.
According to the Instructions for authors:
Main sections must be written in Bold: 2. Materials and Methods
Subsections – in italics, not bolded: 2.1 Life Cycle Assessment (LCA)
Sub- subsections – normal: 2.1.1 Functional units and boundaries definitions
Please adapt the size of Figure 1 to the size of the characters in the main text. In the actual form, it is too small. You can enlarge it on the entire width of the page. Moreover, remove from the Figure GENERAL FLOWCHARTS: it is a single flow chart, and the title of the figure is already mentioned below it.
L202-206. Chose bullets not numbering. The sections are numbered in the same way, so it is confusing so many numberings.
Figure 2. I suggest reducing the width of b and c figures, cutting off/ excluding / removing the part of the graphs with no graphical presentation. Thus, the figures b and c can be placed side by side. The aspect will be more professional.
Under Table 1, please detail all abbreviations used in the Table, as the Instructions for author request: “Acronyms/Abbreviations/Initialisms have been defined the first time they appear in each of three sections: the abstract; the main text; under the first figure or table. When defined for the first time, the acronym/abbreviation/initialism should be added in parentheses after the written-out form”.
Not each phrase is a paragraph. Please merge the paragraphs in more comprehensive ones, as a paragraph is an idea, not a statements/phrase/sentence/ The text will be more “flowy”, easier to follow/understand and the aspect of the paper much neater. The current appearance of the work leaves much to be desired. In order to "sell a product well", in this case to succeed in publishing an article, it is necessary to meet 2 criteria: content and presentation. The text must be clean with no free interspaces, in black colour, more compact, etc.
Figure 3 is blurred. Please provide a better quality one.
Information regarding Materials and Methods, Results and Discussion are mixed. All the graphical part done by the authors is about Results. All the explanations regarding graphical part must be moved in the Discussion.
L364-365. Why Bolded?
L368. Remove the empty line.
Section 4 must be divided in 4. Discussion and 5. Conclusions.
4. Discussion section
L470-513 is not referenced at all. I suggest checking and referring to Bungau et al. Expatiating the impact of anthropogenic aspects and climatic factors on long term soil monitoring and management. Environ Sci. Pollut. Res. 2021, 202, 30528-30550. https://doi.org/10.1007/s11356-021-14127-7 ; Samuel et al. Effects of long term application of organic and mineral fertilizers on soil enzymes. Chim., 69(10), 2018, 2608-2612. https://doi.org/10.37358/RC.18.10.6590
Moreover, it must be emphasized the link between climate changes and fertilisation in orchards: Gitea M.A., et al. Orchard management under the effects of climate change: implications for apple, plum and almond growing. Environ. Sci. Pollut. Res. 2019, 26(10), 9908-9915. https://doi.org/10.1007/s11356-019-04214-1
A paragraph must be also developed regarding the toxicity of the intensive fertilisation, considering the future promising opportunities for agriculture, like nano-technology: Behl et al. The dichotomy of nanotechnology as the cutting edge of agriculture: Nano-farming as an asset versus nanotoxicity, Chemosphere 2022, 288 Part 2, 132533. https://doi.org/10.1016/j.chemosphere.2021.132533
Paragraph L 514-517. Why in blue?
After L517, please add the strengths and the weakness of your study.
L518-525 can be considered the future 5. Conclusions section. Remove “In conclusion” at the beginning of Conclusionssection (L518). It is repetitive and obvious/non-sense.
References must be written in the MDPI style, providing all info requested for references. Please see the Instructions for authors – they are given to be respected, not being optional.
Round 2
Reviewer 3 Report
The authors responded to my requests.